# OpenReview forum: "Securing AI Agents with Information-Flow Control"
_ICML.cc/2026/Conference — Submitted to ICML 2026_

### Official Review · Reviewer_vHsJ · 2026-03-10

**Soundness:** 3
**Presentation:** 2
**Significance:** 3
**Originality:** 3
**Overall Recommendation:** 4
**Confidence:** 5

**Summary:**

This paper studies how to secure AI agents against attacks such as prompt injection using information-flow control (IFC). It presents a formal model to analyze the security of agent planners and enforce confidentially/integrity by (1) information hiding using dual LLM, and (2) information revealing using quarantined LLM. The authors in traduced 4 detailed algorithms, and based on them, the authors propose FIDES, a planner that tracks security labels and enforces security policies. FIDES is evaluated on AgentDojo, and demonstrated significant advantages compared to the baseline.

**Compliance With Llm Reviewing Policy:**

Affirmed.

**Key Questions For Authors:**

Detail comments and questions
===========================

* [Lines 122-133, Algorithm 1]: What is $\sigma$? From later section it means a state, the authors should clarify it when introducing Algorithm 1 here to avoid this kind of confusion. Furthermore, what are $h$ and $r$? From later read, h is the history. The authors should give some high-level intuition to help the reader to understand this algorithm without going forward to later section to figure it out.

* [ Lines 181-191, left column ]:  This description is very off from the presented Algorithm 4 (e,g, there is no request for tool class on line 8, and there is no line 9). Please clarify.

* [Lines 185-186, right column ]:  How expensive is this recursive check? As what authors state in lines 123-124, right column, “ A richer security lattice for confidentiality is the powerset $\mathcal{P}(U)$ of a set of users $U$.” How does this approach scale when the recursive check gets heavy? Would this affect the reported performance on Figure 2 and Figure 3 significantly?

* [Lines 237, left column]:  What is $\pi$ ? From later read, it means a static label. This needs to be introduced more properly

MINOR COMMENTS
=================

* [Line 285, right column] FIDES should be fonted as it was.

**Limitations:**

Nothing notable

**Strengths And Weaknesses:**

STRENGTHS
===========

* This paper presents an interesting connection between AI-agent planner and information-flow control concept such as taint analysis.

* The proposed method is detailly introduced, and the artifact are accessible in the anonymous repository.

* A comparison with relevant baseline approaches is also provided and demonstrated significant advantages of the proposed framework.

* The appendix supports the main text a lot, giving in-depth details on each section about the algorithm, policies, and implementation.


WEAKNESSES
============

* The writing and technical presentation contain some small issues that make the paper difficult to follow on a first read (see detailed comments below).

* The formal model is not introduced rigorously enough, as this is applying formal technique to resolve the problem (see detailed comments below).

* The policies non-interference is introduced (and detailed in appendix C.2), but when discussing the policies used in the experimental results (i.e., Table 2 in Appendix F.1.) it seems there is only functional correctness requirements, without security critical property like NI.

---

> ### Author Rebuttal · Authors · 2026-03-30
>
> Thanks for your detailed comments on presentation. Below, we clarify one of the comments about policies that you listed as a weakness and reply to your key questions.
>
> ### Clarifications on comments
>
>
> > - when discussing the policies used in the experimental results (i.e., Table 2 in Appendix F.1.) it seems there is only functional correctness requirements, without security critical property like NI.
>
> Table 2 in Appendix F.1 lists the concrete security policies we use in the AgentDojo evaluation. We argue in Section 4.5 that these policies, when applied adequately, guarantee non-interference with respect to integrity and explicit secrecy with respect to confidentiality. This considers both the functional behavior of tools (e.g. do they egress data? are they read-only?) and their security implications.
>
> ### Responses to key questions
>
> > 1.  [Lines 122-133, Algorithm 1]: What is $\sigma$? From later section it means a state, the authors should clarify it when introducing Algorithm 1 here to avoid this kind of confusion. Furthermore, what are $h$ and $r$? From later read, $h$ is the history. The authors should give some high-level intuition to help the reader to understand this algorithm without going forward to later section to figure it out.
>
> We include a brief description of Algorithm 1 before it is displayed.  We will elaborate this description and directly refer to the identifiers ($\sigma, h, r$) used in the algorithm for clarity.
>
>
> > 2. [Lines 181-191, left column]: This description is very off from the presented Algorithm 4 (e,g, there is no request for tool class on line 8, and there is no line 9). Please clarify.
>
> Thanks for pointing this out. Line numbers are indeed off due to changes to the layout of the pseudocode not reflected in the writing; we will fix this. Here is a correct mapping:
> - "Lines 4-5" -> "Line 3"
> - "Line 7" -> "Line 5"
> - "Line 8" -> "Line 6"
> - "Line 9" -> "Line 7"
>
>
>  > 3. [Lines 185-186, right column]: How expensive is this recursive check? As what authors state in lines 123-124, right column, “ A richer security lattice for confidentiality is the powerset $\mathcal{P}(U)$ of a set of users $U$.” How does this approach scale when the recursive check gets heavy? Would this affect the reported performance on Figure 2 and Figure 3 significantly?
>
>  This refers to function Hide in Algorithm 7 (lines 1018-1030). The function recurses over structurally smaller values. Each call only needs to compare two labels (line 20 in the algorithm): the cost of this check is independent of the size of the lattice, so there’s no concern scaling this to richer lattices such as powerset lattices of a set of users.
>
>  Figures 2 and 3 report utility results and are not affected by the running time of algorithms. However, we can confirm that the cost of the mentioned check is negligible in our evaluation and on applications that use other label lattices described in the paper.
>
>
> > [Lines 237, left column] What is $\pi$? From later read, it means a static label. This needs to be introduced more properly
>
> $\pi_f, \vec{\pi}$ are indeed a static label and a list of static labels, respectively, as introduced earlier in Section 4.1. We give some examples immediately below that hopefully help clarify the role of these labels. We welcome any concrete pointers to where the current description is lacking and suggestions on how to improve it.

---

> > ### Author Rebuttal · Reviewer_vHsJ · 2026-04-03
> >
> > Thank you for taking the time answering my questions. I still have reservations over the use of functional security policies and labeling them as non-interference (Table 2 in Appendix F). So, I stick with my current score.

---

> > > ### Author Response · Authors · 2026-04-08
> > >
> > > Thank you for your acknowledgement and for the positive score.
> > >
> > > Regarding your remaining concern, in the paper we distinguish between security policies and security properties. A security policy specifies a high-level goal (“this tool must not egress confidential data” or “this tool must not be influenced by untrusted data”). In contrast, a security property specifies the exact sense in which the policy is enforced. For example, when we enforce the policy “this tool must not egress confidential data” in the sense of “weak secrecy”, we guarantee that no sensitive data is copied into the tool arguments. However, “weak secrecy” does not prevent so-called “indirect flows”, which is that the fact that the tool call happens (or not) can already leak one bit of information. This is only achieved by the non-interference property.
> > >
> > > Throughout the paper, we enforce integrity (P-T) in the sense of non-interference (i.e. the adversary must not be able to influence whether a tool call can happen or not) and we enforce confidentiality (P-F) in the sense of weak secrecy (i.e. we allow the adversary to learn whether a tool has been called). This is why in Table 2 we only list which tools are protected by which policy and we leave the properties implicit. We hope this explanation clarifies your concern.

---

### Official Review · Reviewer_ugAa · 2026-03-11

**Soundness:** 4
**Presentation:** 4
**Significance:** 3
**Originality:** 2
**Overall Recommendation:** 3
**Confidence:** 4

**Summary:**

The paper develops a framework and a tool for applying ideas from information-flow control literature in systems security to agentic AI. In particular, the planner keeps track of security labels during tool calling. This provides a rigorous approach to avoid prompt injection attacks. The framework is evaluated on benchmarks from AgentDojo, both in terms of ability to enforce security policies and utility in terms of task completion rate.

**Compliance With Llm Reviewing Policy:**

Affirmed.

**Final Justification:**

the discussion has been substantive. I continue to feel ambivalent about the paper: promising direction but not compelling results. I will keep my score.

**Key Questions For Authors:**

1. Can you compare your results to RTBAS?
2. can you articulate crisply what is the new challenge in agentic AI systems as opposed to more traditional systems with black-box components, in terms of tracking and enforcing security labels?

**Limitations:**

yes

**Strengths And Weaknesses:**

Strengths
+ Information flow control policies is a time-tested technique from security literature. The paper develops a rigorous way of applying it to agentic AI framework, where the planner mediates tool calling and keeps track of state necessary to enforce security policies.
+ The theoretical framework is developed well.

Weakness:
- In the context of extensive work in information flow control policies, I did not find conceptual novelty in the framework. This won't be a concern if the experimental results in agentic AI are significant, but that seems standard, evaluation on one benchmark.
- Evaluation should include more baselines. In particular, RTBAS system seems very relevant, similar methodology, and also evaluated on AgentDojo. Authors do not explain how their method differs and compare in terms of evaluation.

---

> ### Author Rebuttal · Authors · 2026-03-30
>
> Thanks for your review and suggestions to articulate our contributions. We answer key questions below.
>
> > 1.	Can you compare your results to RTBAS?
>
> A key difference between RTBAS and Fides is in how the two systems propagate labels through LLM queries. Fides takes a safe, conservative approach and considers that the output of an LLM query depends on all data in the input prompt. RTBAS takes a less conservative approach based on attention scores or LLM-as-a-judge to approximate how the LLM response depends on data in the prompt. As a consequence, RTBAS suffers from similar limitations to other probabilistic defenses based on input classifiers, i.e., it may miss dependencies on untrusted or confidential data that influence the result. As demonstrated recently (arXiv:2510.09023) with other probabilistic mitigations based on similar methods, adaptive attacks can successfully bypass attention-based and LLM-as-a-judge mechanisms. In the case of RTBAS, a bypass can result in a response influenced by untrusted data labelled as trusted and failure to detect and stop prompt injection attacks.
>
> A quantitative, fair comparison between Fides and RTBAS would require access to an implementation of RTBAS (as far as we know, none is publicly available) and an evaluation against adaptive attacks beyond the AgentDojo baseline attacks using the same labels and security policies. Contrary to Fides' conservative label propagation approach, RTBAS approximative propagation is more likely to be susceptible to adaptive attacks.
>
> > 2.	can you articulate crisply what is the new challenge in agentic AI systems as opposed to more traditional systems with black-box components, in terms of tracking and enforcing security labels?
>
> A straightforward application of information-flow control transposed to agentic AI yields an unacceptable loss of utility. One of the main challenges in building a system like Fides is to make it usable in practice. This requires a careful choice of security policies (we find that non-interference for integrity and explicit secrecy for confidentiality are sensible choices; see Section 4.5), and mechanisms for selectively hiding and revealing information from the planner (for which we adapt and extend the Dual LLM pattern with constrained decoding). Dynamic taint-tracking and enforcing security policies in agentic AI systems is mostly analogous to traditional systems. In this sense, there are no sufficiently robust techniques or architectural designs that can more precisely track information flow through LLM queries.

---

> > ### Author Rebuttal · Reviewer_ugAa · 2026-04-01
> >
> > Thank you for clear answers to my questions. This is certainly a promising direction, but as explained in "weaknesses" part of my review, given that the technique is based on security literature, evaluation of the ideas in the agentic framework could/should use a more challenging case study, IMO, so I will keep my score.

---

> > > ### Author Response · Authors · 2026-04-08
> > >
> > > > This is certainly a promising direction, but as explained in "weaknesses" part of my review, given that the technique is based on security literature, evaluation of the ideas in the agentic framework could/should use a more challenging case study, IMO, so I will keep my score.
> > >
> > > We chose AgentDojo because it is an established benchmark and facilitates comparison with related work. AgentDojo represents better than other benchmarks the complexities of practical agentic systems. We considered but dismissed ASB (ICLR’25) [1] and InjecAgent (ACL'24) [2] as unrealistic. ASB does not use models’ tool-calling capabilities. Instead, it simulates tools that do not take any parameters and that return textual descriptions about what a real tool would do. Unlike AgentDojo, InjecAgent comprises single-turn tasks that do not fully exercise planning capabilities and information-flow tracking across turns.
> > >
> > > We additionally evaluated FIDES on WASP (NeurIPS'25) [3]. The benchmark evaluates browser-use agents’ robustness to prompt injection attacks on the VisualWebArena benchmark using simulated Reddit and GitLab environments. It includes 21 injection tasks embedded in posts or issues, measuring attack success and task completion rate. We observe that FIDES blocks all attacks and achieves a higher task completion rate compared to the Basic planner. This is expected by design of FIDES as the Basic planner often gets confused by injected instructions in its context whereas the injected instructions are hidden behind variables for FIDES, preventing the context from being tainted. We hope that this addresses your concern and if so, please consider adjusting your score accordingly. We will add these results to the paper.
> > >
> > > | Model   | Environment | ASR Basic (%) | ASR FIDES (%) | TCR Basic (%) | TCR FIDES (%) |
> > > | ------- | ----------- | ------------- | ------------- | ------------- | ------------- |
> > > | GPT-4o  | GitLab      | 20.80         | 0             | 64.10         | 75.00         |
> > > | GPT-4o  | Reddit      | 47.20         | 0             | 36.10         | 55.60         |
> > > | o1      | GitLab      | 29.20         | 0             | 62.50         | 85.40         |
> > > | o1      | Reddit      | 36.10         | 0             | 47.20         | 50.00         |
> > > | o4-mini | GitLab      | 25.00         | 0             | 64.60         | 72.90         |
> > > | o4-mini | Reddit      | 52.80         | 0             | 36.10         | 63.90         |
> > >
> > > [1] Agent Security Bench (ASB): Formalizing and Benchmarking Attacks and Defenses in {LLM}-based Agents, Anrong Zhang and Jingyuan Huang and Kai Mei and Yifei Yao and Zhenting Wang and Chenlu Zhan and Hongwei Wang and Yongfeng Zhang, https://openreview.net/forum?id=V4y0CpX4hK [ICLR’25]
> > >
> > > [2]  InjecAgent: Benchmarking Indirect Prompt Injections in Tool-Integrated Large Language Model Agents, Qiusi Zhan and Zhixiang Liang and Zifan Ying and Daniel Kang, https://aclanthology.org/2024.findings-acl.624/ [ACL’24]
> > >
> > > [3] WASP: Benchmarking Web Agent Security Against Prompt Injection Attacks, Ivan Evtimov and Arman Zharmagambetov and Aaron Grattafiori and Chuan Guo and Kamalika Chaudhuri, https://arxiv.org/abs/2504.18575 [NeurIPS’25]

---

### Official Review · Reviewer_s8c4 · 2026-03-12

**Soundness:** 3
**Presentation:** 2
**Significance:** 3
**Originality:** 2
**Overall Recommendation:** 4
**Confidence:** 2

**Summary:**

I am from a different field from this paper (although I am familiar with taint analysis/data flow analysis), so my understanding may be lacking, so apologies in advance for any misunderstandings.

This paper uses ideas from taint analysis and information flow to provide better security against prompt injection attacks. The system is called FIDES. They start out by labeling untrusted sources of data (usually the results of tool calls), and then whenever that flows through as a potential input to an LLM agent, it is hidden behind a variable. The LLM only sees the variable at first. For example, if an LLM calls a tool call to read an email response, it is hidden behind a variable `#email_response`. Then, when it wants to read the result of the tool call, the variable goes through a quarantining LLM which filters the results using constrained decoding, limiting the possible effect of attacker interference (although probably not completely negating it). They show improved results on AgentDojo, stopping all prompt injection attacks.

**Compliance With Llm Reviewing Policy:**

Affirmed.

**Final Justification:**

I am generally positive on this paper, although I am not familiar with the field. It's a well-motivated problem, and it seems like their approach notably improves performance.

I mainly had a few reservations on the presentation, since I felt it was overly complicated. However, the authors say they will address this through the rebuttal, and I don't think it's a big enough issue to prevent the paper from being accepted.

**Key Questions For Authors:**

Can the authors confirm my understanding of the paper? If my summary is correct, I have two other questions:

- Line 293 states that you had to label all instances. How precise do the labels have to be? I saw in the appendix that you could do something simple like label all external things as untrusted. But in practice and in the evaluation, how precise are they?
- Does constrained decoding lose too much information? If I understand, the quarantining LLM uses constrained decoding which filters the results to a primitive. Does this lose too much information?

**Limitations:**

yes

**Strengths And Weaknesses:**

#### Strengths
It seems like a nice approach from taint analysis, and their results are promising. It makes sense to have additional guardrails around potentially untrusted input, and the results of certain tool calls should definitely be examined more closely.


#### Weaknesses
if my summary above is correct, I think the main criticism I have of this paper is it's presentation. I feel like the idea is quite straightforward and intuitive, but the methodology (section 3) feels extremely bloated. I personally feel like four algorithms and the other notation is a bit overkill. I also think that Figure 1 doesn't do a good job explaining the idea. There's nothing in there that really implies information flow to me. At the very least, I think this paper could use an "overview" section that explains the idea at a higher and more intuitive level.

---

> ### Author Rebuttal · Authors · 2026-03-30
>
> Thanks for your honest feedback and your suggestions to improve the presentation. We clarify some possible misunderstandings in your summary, discuss your comments on the presentation, and answer to your key questions below. Please feel free to ask any follow up questions you may have.
> ### Clarifications about the summary in the review
>
> - We consider both data integrity **and** confidentiality. So, the analysis about untrusted data extends to confidential data. In general, only data that would lower the integrity or raise the confidentiality of the planner's context is hidden in variables.
> - The planner can pass variables in arguments to tool calls, including to the built-in Quarantined LLM  tool ($\texttt{query-llm}$). The results of $\texttt{query-llm}$ follow the same logic as other tools and are placed in variables if they would taint the planner's context. At any time, the planner can decide to reveal the content of a variable using a separate $\texttt{inspect}$ tool (see Section 4.4).
> ### Comments about presentation
>
> To improve the presentation, we will summarize selected content from the first 3 sections in an Overview section at the end of the introduction, describing our techniques and the intuition behind them.
>
> Because Figure 1 appears early in the paper, we kept it schematic. The arrows in the diagram indicate flow of information, and the label icons on them show where information-flow labels are passed and propagated.  We welcome suggestions to improve the figure. For instance, we considered adding an example of a user query-response flow through the diagram, with numbers over arrows indicating the order of communication. Would you consider this useful?
>
> ### Responses to key questions
>
> > 1. Line 293 states that you had to label all instances. How precise do the labels have to be? I saw in the appendix that you could do something simple like label all external things as untrusted. But in practice and in the evaluation, how precise are they?
>
> In our evaluation on AgentDojo we take a conservative approach: if there is at least one attack in the benchmark where a field is used in a prompt injection attack, we label of data across the benchmark in the same field as untrusted (see Appendix F.1 for more details). For instance, email bodies are labeled as untrusted. In practice, when such ground truth may not be available, we leverage existing metadata in real world applications (see lines 643-650). For instance, the subject and body of emails internal to an organization may be labelled as trusted and confidential, whereas emails originating from outside the organization can be labelled as untrusted. Coarse labels are often sufficient (e.g. we did not see a need to label nested fields in our evaluation; see lines 977-983). There is of course a trade-off between label granularity and expressiveness, and different applications can benefit from more fine-grained labels.
>
> >2. Does constrained decoding lose too much information? If I understand, the quarantining LLM uses constrained decoding which filters the results to a primitive. Does this lose too much information?
>
> We use constrained decoding to enforce an output schema in Quarantined LLM queries ($\texttt{query-llm}$ tool). This schema can be chosen by the planner without restrictions, so it does not necessarily lose information. This is used in plans to restructure or extract information to pass results to other tool calls, something that does not require enforcing any constraints on the amount of information. We do discuss extending labels to allow a planner to reveal the value of variables with restricted schemas without tainting its context (see Appendix B.2). In this case, information loss is a necessary justification.

---

> > ### Author Rebuttal · Reviewer_s8c4 · 2026-03-31
> >
> > I see. I think with regards to figure 1 that would definitely be helpful (although it's hard to say without seeing it). I feel like what's confusing to me is that it's not clear where the "entry point" is which is what I usually associate with an information flow diagram.
> >
> > But overall, this seems like a good paper to me. I'm keeping my score (I don't feel comfortable raising my score to an "accept" given my unfamiliarity with the field and related work).

---

> > > ### Author Response · Authors · 2026-04-08
> > >
> > > Thanks again for your suggestions to improve Figure 1. Our revised diagram and accompanying example will make clear where the entry point is (the user submitting a task to the agent loop) and the concrete order in which information flows through the diagram. We will complement Figure 1 with a sequence diagram to visualize this explicitly.
> > > Please consider adjusting your Presentation score if you believe that our rebuttal and the changes we suggest address your concerns in this respect.

---

### Official Review · Reviewer_guAi · 2026-03-15

**Soundness:** 4
**Presentation:** 3
**Significance:** 3
**Originality:** 3
**Overall Recommendation:** 4
**Confidence:** 2

**Summary:**

The paper presents FIDES, which uses information flow control to provide security guarantees against indirect prompt injection attacks. The core idea is to attach integrity and confidentiality labels to all data, hide untrusted tool results, and use a quarantined LLM with constrained decoding to inspect data when needed. The method blocks prompt objections while performing more tasks than a simple IFC planner baseline.

**Compliance With Llm Reviewing Policy:**

Affirmed.

**Final Justification:**

I think this is a borderline paper and have decided to stick to my positive score due to my lack of familiarity with the area.

**Key Questions For Authors:**

- How might FIDES be optimized to reduce latency without sacrificing its deterministic security guarantees?

- In scenarios where upstream data sources provide no reliable metadata or structural hints, how do you envision FIDES preventing utility from dropping especially when it is forced to label external inputs as untrusted?

- For constrained outputs like booleans or enums, an adversary can still semantically manipulate the result (e.g., forcing a false ‘true’ flag for some task). Since these values drive subsequent planning decisions while remaining policy-compliant, how do you bound the damage from semantically incorrect but schema-valid constrained outputs?

**Limitations:**

Yes

**Strengths And Weaknesses:**

Strengths

- The paper separates the planning loop from the planner, which makes the system modular. The characterization for integrity and confidentiality is well motivated. This is an improvement over previous IFC work for AI agents.

- The classification of tasks is useful and provides a principled framework for reasoning about what an IFC-enforcing planner can and cannot accomplish. It allows the paper to make the security-utility tradeoffs clear so as to motivate FIDES. There is also a human oracle baseline present, which strengthens the results.

- There are several useful insights in the paper (for e.g. only tools with labels more restrictive than the current context need to be hidden) and it significantly outperforms the rigid Dual LLM / Variable Passing approach while maintaining deterministic guarantees.

Weaknesses

- The labeling method is hard to use in the real world. Currently labels are assigned by inspecting which data fields  are targeted by AgentDojo’s injection tasks (see Section 6) but this information about the attack surface may not be available in practice.

- Relying on a quarantined LLM to inspect variables inherently increases the number of turns required to complete tasks. The authors acknowledge a 2-3x increase in token utilization compared to a basic planner.

---

> ### Author Rebuttal · Authors · 2026-03-30
>
> Thank you for your detailed feedback. We provide brief answers to your key questions below. We will be glad to provide any further clarifications if needed.
>
> > 1. How might FIDES be optimized to reduce latency without sacrificing its deterministic security guarantees?
>
> In this paper, we focus on evaluating the impact of information-flow control on utility and security. We deliberately leave efficiency optimizations to future work. To answer the question, we see several ways to improve efficiency:
>
> - We use off-the-shelf models and instruct them on how to work with quarantined LLM queries and variables, a task they have not been explicitly trained for. We believe that fine-tuning models to use these mechanisms effectively can unlock additional gains in utility and efficiency.
>
> - Giving the planner’s LLM access to the enforced policies and dynamic labels attached to tool results can help it generate plans that anticipate and avoid policy violations, reducing the need for costly extra turns to recover (cf. Appendix G, paragraph “Policy-aware planners”).
>
> - Static planning, where the system generates a plan based on the user’s task in the first turn and follows it in subsequent turns, drastically reduces latency and token usage. However, this comes at the cost of reduced ability to solve some data-dependent tasks. Hybrid solutions that revisit an initial plan when runtime failures occur may strike a better balance between latency and expressivity.
>
> > 2. In scenarios where upstream data sources provide no reliable metadata or structural hints, how do you envision FIDES preventing utility from dropping especially when it is forced to label external inputs as untrusted?
>
> Indeed, when reliable labels are unavailable, labelling data required for planning conservatively as "confidential" or "untrusted" inevitably degrade the utility of IFC-based defenses. However, our experience indicates that real-world applications often include metadata, structural or semantic hints that suffice to infer reliable labels, a point that we discuss in Appendix B.1. A pragmatic solution to increase utility is to leverage existing AI-based classifiers to infer labels. Although these classifiers are fallible, they can be surgically applied where labels cannot be inferred by other means and with conservative thresholds that minimize false negatives. See Appendix G (paragraph “Dynamic and Fine-grained Policies”) for a brief discussion.
>
> > 3. For constrained outputs like booleans or enums, an adversary can still semantically manipulate the result (e.g., forcing a false ‘true’ flag for some task). Since these values drive subsequent planning decisions while remaining policy-compliant, how do you bound the damage from semantically incorrect but schema-valid constrained outputs?
>
> Appendix B.2. shows how to use labels to track the amount of information flowing from values with limited capacity (e.g. booleans) into a planner’s context. This allows developers to specify more permissive policies that keep control over the influence of untrusted or confidential data on planning decisions. The reviewer is right that such policies open the door for possible attacks. That is why our experimental evaluation on AgentDojo does not use such permissive policies and satisfies strict non-interference for integrity and explicit secrecy for confidentiality.
>
> In practice, permitting the flow of a limited amount of confidential information into the planner’s context can be a sensible compromise. This guarantees bounds on the amount of data that an attack may exfiltrate.

---

> > ### Author Rebuttal · Reviewer_guAi · 2026-04-03
> >
> > I thank the authors for their response and maintain my positive score. Similar to reviewer s8c4, I am not entirely familiar with the subfield.

---

### Decision · Program_Chairs · 2026-04-30

**Decision:**

Reject

**Comment:**

The paper considers the use of information-flow control (IFC) to secure AI agents against prompt injection attacks. In particular, the paper considers how to use IFC to develop planners that orchestrate calls to LLMs and other tools securely.

The rebuttal clarifies that a fair comparison with other baselines, such as RTBAS, can be challenging, particularly due to the lack of public implementation. This paper makes the implementation of the proposed approach public, a strong point of this manuscript, as noted by `vHsJ`.

The rebuttal also notes that probabilistic defenses may fail against adaptive attacks, whereas the proposed approach is deterministic. However, such adaptive attacks are not in the scope of the paper. This demonstrates that the suggestion from `ugAa` to use a more challenging case study can be beneficial.

The technical presentation had some minor inconsistencies that were clarified during the rebuttal; nevertheless, as `vHsJ` notes, the formal model is not described rigorously. Furthermore, `s8c4` found the paper's presentation could be more streamlined. Given that one of the paper's core contributions is the development of a formal model to control information flow in planners, the presentation is a major issue.

In short, this paper presents a method for securing AI agents, supported by empirical evidence. Nevertheless, its presentation needs considerable improvements to do justice to its conceptual contributions.  Therefore, I recommend rejecting this paper.